# UCH-L1 in Alzheimer’s Disease: A Crucial Player in Dementia-Associated Mechanisms

**DOI:** 10.3390/ijms26189012

**Published:** 2025-09-16

**Authors:** Elisa Porchietto, Giulia Morello, Giulia Cicilese, Innocenzo Rainero, Elisa Rubino, Elena Tamagno, Silvia Boschi, Michela Guglielmotto

**Affiliations:** 1Department of Neuroscience, University of Torino, Via Cherasco 15, 10126 Torino, Italy; elisa.porchietto@unito.it (E.P.); g.morello@unito.it (G.M.); giulia.cicilese@unito.it (G.C.); innocenzo.rainero@unito.it (I.R.); elisa.rubino@unito.it (E.R.); elena.tamagno@unito.it (E.T.); michela.guglielmotto@unito.it (M.G.); 2Neuroscience Institute of Cavalieri Ottolenghi Foundation (NICO), University of Torino, Regione Gonzole 10, Orbassano, 10043 Torino, Italy

**Keywords:** UCH-L1, amyloid-β, tau hyperphosphorylation, synaptic plasticity, neuroinflammation, biomarker, protein misfolding, Alzheimer’s disease (AD)

## Abstract

Ubiquitin carboxyl-terminal hydrolase L1 (UCH-L1) is a critical deubiquitinating enzyme that is highly expressed in the central nervous system, where it participates in protein degradation and turnover as part of the ubiquitin–proteasome system (UPS). Convincing evidence supports the role of UCH-L1 dysfunction in several neurodegenerative disorders, given its unique position at the crossroad of several aetiopathogenic pathways, including those implicated in Alzheimer’s disease (AD) onset. Indeed, UCH-L1 depletion correlates with decreased levels of triggering receptor expressed on myeloid cells 2 (TREM2), with consequent effects on neuroinflammation. Notably, UCH-L1 can affect the level of phosphorylated tau protein, thus contributing to the formation of neurofibrillary tangles (NFTs). In addition, UCH-L1 influences β-Secretase 1 (BACE1) expression, resulting in the abnormal accumulation of amyloid-β plaques in brain parenchyma. These findings underline UCH-L1’s centrality in maintaining the homeostasis of protein folding and aggregation, which are significantly impaired in AD and AD-related dementias. Given these assumptions, UCH-L1 is recognized as a potential biomarker for AD, highlighting its relevance in governing the fate of crucial pathological mediators of cognitive impairment and neurodegeneration. Herein, we contextualize the involvement of UCH-L1 in different dementia-associated pathways and summarize the state of the art of UCH-L1 as a biomarker for AD diagnosis.

## 1. Mechanistic Insights into UCH-L1 Enzyme Structure and Function

Ubiquitination is a crucial post-translational modification that regulates protein fate by modulating stability, turnover, and overall functionality [1]. This phenomenon is tightly regulated by an enzymatic cascade, which ultimately leads to the conjugation of ubiquitin to lysine residues on the target protein, resulting in either mono- or polyubiquitinated modifications [2]. A distinctive feature of this process is its reversibility, which is guaranteed by deubiquitylating enzymes (DUBs). These proteases can remove ubiquitin from substrate proteins, edit ubiquitin chains, and process its precursors [3]. More than 100 DUBs are recognized in the human genome and classified in distinct families, according to their sequence and structural similarities of the catalytic domain, namely metalloproteases and cysteine proteases [4,5]. Among the latter, the ubiquitin C-terminal hydrolase (UCH) sub-family includes four members, UCH-L1, UCH-L3, UCH-L5, and BRCA1 associated protein 1 (BAP1). UCH-L1 is predominantly expressed in the central nervous system (CNS), representing 1–5% of soluble brain proteins [6]. Its expression is limited in peripheral tissue such as the gonads, but it is upregulated in different cancers [7]. In situ hybridization data of mouse brains showed an almost ubiquitarian expression of UCH-L1, and this is true in rat brains as well. In detail, the robust positive UCH-L1 immunocytochemical (ICC) staining of murine brains was documented in different subregions, i.e., the cortex, hippocampus (dentate gyrus and CA1 layer), thalamus, hypothalamus, brain stem, cerebellum, and corpus callosum [8]. These preclinical data were also paralleled by an investigation on the human brain, where UCH-L1 concentration was slightly lower in a white matter area, compared with a gray one, such as the cortex, caudate nucleus, and hippocampus; moreover, the concentration of the enzyme in the internal capsule, thalamus, brainstem, and cervical spinal cord was not different when compared to the cortical regions. In this study, the most abundant content was detected in the hippocampus, while the lowest was in the cerebellum [9].

The UCH-L1 enzyme consists of 223 amino acids and has a molecular weight of ~24.8 kDa [6,10]. UCH-L1 is a globular protein characterized by a “Gordian knot”-like structure, from which a conserved N-terminal C12 peptidase catalytic domain originates, containing cysteine, histidine, and aspartate aminoacidic residues, which are responsible for hydrolytic activity. By contrast, the C-terminal extension contributes to the protein solubility and ubiquitin binding, while an unstructured loop represents the size-limiting gate for the proteins that UCH-L1 can process [11,12,13,14,15]. Despite the fact that UCH-L1 is prevalently soluble in the cytosolic compartment (UCH-L1S), around 20–50% could be membrane-associated (UCH-L1M) [13,16], and the latter has been associated with aberrant α-synuclein accumulation in Parkinson’s disease (PD) [17]. Interestingly, proteome analysis of post-mortem brain tissues with Alzheimer’s disease (AD) suggests an altered UCH-L1M accumulation [18], showing that the further investigation of the sub-localization of this enzyme could better elucidate the crucial role of UCH-L1 in neurodegenerative disorders.

As extensively reviewed by Mi and Graham [19], UCH-L1 can undergo different post-transcriptional modifications at different sites, potentially affecting its enzymatic function. Indeed, UCH-L1 oxidation has been documented in the brain of AD patients, namely the formation of carbonyls and methionine/cysteine oxidations. These modifications lead to oxidative damage, decreased enzymatic activity, and the consequent accumulation of misfolded proteins [20,21,22,23]. Moreover, UCH-L1 is less soluble in its oxidized form and increases proportionally to the number of tau-immunoreactive tangles [11]. S-nitrosylation is another post-transcriptional modification documented in AD patients: a transnitrosylation cascade promotes the transfer of a nitric oxide (NO) group from UCH-L1 to Cyclin-dependent kinase 5 (Cdk5) and subsequently to Dynamin-related Protein 1 (Drp1) protein, ultimately contributing to synaptic loss [24]. Oxidized and nitrosylated forms of UCH-L1 could represent a double-edged sword: on one hand, these post-transcriptional modifications exert detrimental effects by perpetuating oxidative damage, promoting synaptic loss, and ultimately impairing UCH-L1 homeostatic functionality; on the other hand, they may constitute potential targets to allow the selective suppression of the modified UCH-L1isoforms, without affecting the expression levels or the physiological activity of the native protein. Farnesylation and other lipid modifications have also been reported to affect UCH-L1 function and have been implicated in PD [17,25]. In vitro data on the human neuroblastoma cell line (SH-SY5Y) suggest that S-mercuration could also impair UCH-L1 activity and reduce the levels of monoubiquitin, thus compromising UCH-L1 activity [26].

UCH-L1 is not only subjected to different post-transcriptional modifications, but different polymorphisms have also been described. For instance, S18Y polymorphism was believed to be protective against AD [27,28]; nevertheless Zetterberg and colleagues showed the lack of positive association in a larger cohort of patients [29].

Despite its well-recognized activity as a hydrolase, an additional important function has been discovered, hinting at the pivotal role of this mediator in governing cellular homeostasis. Indeed, Liu and colleagues identified in vitro dimerization-dependent ubiquityl ligase activity. This enzymatic process mediates the linkage of a Lys-63-linked ubiquitin chain to substrates, preventing their proteasomal degradation and thereby promoting their stabilization. This process seems to support the accumulation of α-synuclein in PD [30] and interestingly, in its ligase activity, UCH-L1 does not require ATP to activate free ubiquitin for its conjugation to substrates, highlighting a unique characteristic that may also influence its targeting [31]. Despite the appealing implications of a dual role for the UCH-L1 enzyme, its in vivo ligase activity requires further investigations. In particular, at least in the CNS, it is still unclear whether the enzyme undergoes dimerization under physiological conditions, or what the precise nature of its ligation substrates might be [32]. Addressing these questions could help to dissect the multifaceted role of UCH-L1 in the brain, both in physiological and pathological contexts [33].

On the other hand, UCH-L1 acts as a monoubiquitin stabilizer, influencing ubiquitin directly by maintaining a stable pool of mono-ubiquitin, which is essential for the proper function of protein degradation machinery [14]. Moreover, UCH-L1 elongates ubiquitin half-life, confirming its crucial role in the homeostasis of this protein [34]. Consistently, UCH-L1 can slowly cleave ubiquitin gene products in vitro as well, to generate free monomeric ubiquitin from ubiquitin proproteins [35]. When ubiquitin is present in a defective variant, namely ubiquitin B + 1 (UBB + 1), it can compete with ubiquitin for UCH-L1, leading to the aberrant accumulation of amyloid precursor protein (APP) and affecting amyloid-β (Aβ) plaque processing. Not by chance, UBB + 1 is the result of a non-hereditary RNA frameshift mutation, which is detectable in AD patient brains [36] already at the early stage of AD. Of particular note, UBB + 1 can evoke Aβ deposition and insoluble hyperphosphorylated tau aggregates in a 3D human neural culture, while silencing UBB + 1 expression counteracts AD hallmarks in this model system [37].

UCH-L1 is a crucial enzyme in brain homeostasis, and this is supported by the fact that alterations in its functionality result in pathological outcomes: this review aims to dissect the role of UCH-L1 in neurodegenerative disorders, with a particular focus on AD. Moreover, it provides a comprehensive overview of the most relevant clinical data, with the ultimate goal of evaluating the role of UCH-L1 as a candidate biomarker in neurodegenerative disorders.

## 2. Unveiling the Role of UCH-L1 in Governing Crucial AD-Associated Hallmarks

Alzheimer’s disease, the most common form of dementia worldwide, is a multifactorial disorder characterized by an intricate interplay among Aβ plaque deposition, tau-mediated neurofibrillary tangle formation (NFT), and neuroinflammation, which result in synaptic alterations and ultimately neurodegeneration [38]. In this complex framework, UCH-L1 presents multiple points of connection with different key drivers of AD aetiopathogenesis. In this paragraph, we explore the relationship between this UPS enzyme and crucial histopathological hallmarks of AD, namely Aβ, tau protein, and NFTs, and with synaptic loss.

### 2.1. Amyloid-β Plaque Deposition

Consistent scientific reports have unraveled the effects of UCH-L1 on Aβ plaque synthesis and accumulation [39,40]. Indeed, UCH-L1 interacts directly with the amyloid precursor protein (APP): Zhang and colleagues demonstrated the co-immunoprecipitation of APP and UCH-L1 in Haw cells as well as UCH-L1-driven APP ubiquitination, which lead to APP lysosomal degradation, thus proposing a protective role exerted by UCH-L1 in promoting the clearance of APP [40]. The relevance of UCH-L1’s protective effects is proven by the fact that it is reduced in post-mortem brains with AD [21,41,42], even if conflictual reports have been documented [18]. Despite this, a recent study from Toyama and colleagues reported UCH-L1 redistribution in the gray matter of AD patients in the area characterized by Aβ plaque deposition [43], suggesting a tight correlation between UCH-L1 and Aβ plaque. Moreover, a mouse model of gracile axonal dystrophy (gad), which resulted in the lackof UCH-L1 because of the spontaneous deletions of exons 7 and 8, presents the aberrant accumulation of Aβ in the ascending gracile tract [44], strengthening the concept of UCH-L1 dysfunction in the onset of AD. By contrast, when APP23/PS45 transgenic mice are treated with AAV1-UCH-L1-GFP, a UCH-L1-expressing rAAV, there is a reduction in plaque formation, suggesting that restoring UCH-L1 activity ameliorates this AD pathological hallmark [40]. The protective effect of exogenous UCH-L1 was already discussed by Gong et al., who used a TAT-linked UCH-L1 to repristinate its enzymatic functionality and synaptic homeostasis, both in Aβ-oligomer-treated hippocampal slices from wild-type animals and in transgenic APP/PS1 mice [39].

UCH-L1 can also indirectly affect the fate of Aβ plaques acting on BACE1 (β-site amyloid precursor protein cleaving enzyme 1). Indeed, BACE1 is the rate-limiting enzyme for the amyloidogenic cutting of APP. In particular, the proteolytic action of BACE1 generates a soluble APP β fragment and a C99 transmembrane terminal. The C99 fragment endures an intra-membranous cut by γ-secretase, generating the Aβ oligomers, which exert deleterious effects [45,46]. BACE1 is usually degraded via the ubiquitin–proteasome (UPS) pathway and the pharmacological blockade of UPS with lactacystin results in the inhibition of BACE1 degradation in a time- and dose-dependent manner [47]. The first functional link between UCH-L1 and BACE1 was established in vitro using HEK293 cells overexpressing UCH-L1: BACE1 half-life was reduced and most notably, the overexpression of UCH-L1 was responsible for the reduction in the production of C99 fragments and Aβ levels. These results highlight UCH-L1’s role in the orchestration of BACE1 activity: indeed, when inhibited, UCH-L1 significantly increased BACE1 protein levels, thus indirectly affecting APP processing too. Moreover, for the in vivo investigation of the UCH-L1-null gad mice, a robust increase in endogenous BACE1, C99 fragment, and Aβ levels was documented [48]. Remarkably, parallel works of our research group investigated further crucial mechanisms potentially involved in the UCH-L1 downregulation and consequent BACE1 upregulation. We confirmed that an increase in BACE1 expression was associated with a reduction in UCH-L1 expression and activity in different cellular models in the presence of Aβ1-42 as a noxious stimulus. Notably, this event was triggered by a master regulator of cell fate, the Nuclear Factor kappa B (NF-κB) pathway. Indeed, the pharmacological targeting of this cascade rescued the Aβ-42-driven decrease in UCH-L1 and the consequent BACE1 increased expression [41]. All of these results were subsequently confirmed in a 5xFAD mouse model, where the use of a UCH-L1 restoration peptide seems to counteract BACE1 induction and prevent cell death in vivo [49].

In conclusion, UCH-L1 regulates Aβ plaque formation through both direct mechanisms, via APP ubiquitination and degradation, and indirect mechanisms, by modulating BACE1 stability and activity. Notably, decreased UCH-L1 expression and activity lead to a reduction in the free ubiquitin pool and, ultimately, impair proteasomal protein degradation, including BACE1, in both preclinical and clinical models of AD (Figure 1). Intriguingly, we further observed that the effects of UCH-L1 on BACE1 are also mediated by the crucial relationship between UCH-L1 and NF-κB. Indeed, this transcription factor, once activated, suppresses UCH-L1 gene expression [50]. These findings highlight a crucial role for UCH-L1 in regulating the multiple cellular pathways implicated in the onset and progression of AD.

### 2.2. Tau Protein

The tau protein, encoded by the MAPT gene (Microtubule-Associated Protein Tau), has the primary role of promoting tubulin assembly and stabilizing polymerized microtubules [51]. This protein is highly enriched at the axonal level, where it is crucial for the homeostasis of axonal microtubules and necessary for cytoskeleton integrity and neuronal shape [52]. Tau efficiency is tightly regulated by different post-translational modifications, in particular phosphorylation, which is the result of a crucial balanced mechanism governed by phosphatases and kinases. Specifically, the addition of a phosphate group on different tau serine aminoacidic residues decreases tau affinity for microtubules and promotes protein aggregation [53]. Notably, on the one hand, hyperphosphorylated tau shows a decreased affinity for microtubules, thus leading to tau detachment and misfolding, on the other hand, its hyperphosphorylation tends to promote tau–tau affinity [54]. Hence, in the earlier steps, tau forms paired helix filaments (PHFs), which evolve in intracellular protein aggregates, the neurofibrillary tangles (NFTs). NFTs lead to an abnormal loss of communication between neurons and signal processing, eventually leading to apoptosis [55]. A dual connection between tau pathologically phosphorylated and the activity of the proteasome system exists: if the proteasome system is crucial for aggregated tau removal, as the tauopathy proceeds, tau itself starts to exert a neurotoxic effect on UPS [56].

Several lines of evidence suggest that UCH-L1 downregulation, which is documented in an AD pathological context, is also related to the tau protein and NFTs [21,57,58,59]. The pharmacological inhibition or siRNA treatment of UCH-L1 results in a decreased microtubule-binding ability and increased phosphorylation status of tau protein in Neuro2a (N2a) cells and HEK293/tau441 cells, respectively. Tau abnormal aggregation and ubiquitination of the protein following UCH-L1 inhibition have also been detected [57]. These findings are suggestive of UCH-L1’s crucial position in governing tau protein homeostasis, and of how UCH-L1 dysfunction may affect the degradation rate of ubiquitinated and hyperphosphorylated tau. Moreover, the siRNA-mediated knockdown of endogenous UCH-L1 in SH-SY5Y cells resulted in an increased phosphorylation status of tau on Ser 396, which was, on the contrary, decreased when UCH-L1 was overexpressed [60]. Dysregulated UCH-L1 is also associated with the dysfunctional aggresome, a central player for the correct clearance of intracellular toxic protein aggregates such as tau. Indeed, Yu and colleagues observed that UCH-L1 inhibition can impair aggresome formation by downregulating protein histone deacetylase 6 (HDAC6) activity in HEK293 stably expressing tau441 [59]. UCH-L1 co-localisation in the NFTs of AD patients and the inverse correlation between soluble UCH-L1 and NFTs are convincingly consolidated results widely accepted in the scientific community throughout decades of scientific evidence [21,58]. Of particular note, an altered distribution of UCH-L1 between soluble and particulate brain fractions in AD-affected cortexes with respect to controls has been documented, which is consistent with UCH-L1 sequestration into NFTs and plaques [61].

Altogether, the strict correlation between UCH-L1 and tau protein, a crucial histopathological feature of AD, corroborates the multifaced interplays in which UCH-L1 is involved, highlighting its importance in the onset of the pathological phosphorylation of tau and ultimate accumulation within the NFTs (Figure 2).

### 2.3. Synaptic Loss

The impairment of synaptic plasticity is a critical downstream consequence of the pathological accumulation of Aβ plaques and NFTs in AD, which ultimately leads to cognitive decline and the impairment of memory [62]. As consistently reported in the previous paragraphs, a reduced ubiquitin pool and decreased UCH-L1-mediated protein degradation disrupt key cellular processes, such as Aβ plaque and tau protein regulation, with the impairment of synaptic circuitry and compromission of learning and memory. The first evidence of UCH-L1’s interference with memory formation was carried out by Hedge et al., who showed the crucial role of UCH-L1 in promoting the degradation of the regulatory subunit of protein kinase A (PKA) in the Aplysia model [63]. Indeed, PKA phosphorylates the transcription factor CREB (cyclic AMP–responsive element–binding protein), enhancing synaptic functionality and eventually affecting memory formation [64]. The critical effects of UCH-L1 on PKA-CREB cascade was further investigated in a transgenic mouse model of AD: in this case, restoring UCH-L1 homeostatic functionality counteracts the Aβ-induced inhibition of long-term potentiation (LTP), rescuing basal neurotransmission and synaptic plasticity and boosting associative memory in APP/PS1 mice. The authors convincingly showed that treatment with TAT-linked UCH-L1 repristinated normal levels of the PKA-regulatory subunit IIα PKA activity and CREB phosphorylation [39]. In line with this, Sakurai et al. observed a reduction in memory in passive avoidance learning, exploratory behavior, and ultimately synaptic plasticity in gad mice. These alterations were ascribable to impaired CREB phosphorylation, which was not maintained functional over time [65]. Moreover, Zhang and colleagues again confirmed the beneficial effects of UCH-L1 overexpression, not only in reducing Aβ and NFTs burdens, but most notably in improving memory deficits in APP23/PS45 transgenic mice. This effect however was not confirmed in the APP23/gad mice, suggesting that the protection of exogenous UCH-L1 cannot be obtained in the presence of a severe loss-of-function mutation UCH-L1 [40]. Of particular note, the stimulation of synaptic activity by an N-methyl-D-aspartate (NMDA) receptor significantly upregulates UCH-L1 activity in hippocampal neurons and promotes an increase in free monomeric ubiquitin, suggesting a connection with the NMDAR cascade, which can improve the availability of ubiquitin to support proteasome activity and consequently synaptic function. Conversely, the pharmacological blockade of UCH-L1 impairs ubiquitin levels and alters synaptic protein clusters, spine morphology, and density. These results suggest a significant role of UCH-L1 in synaptic remodeling by modulating free ubiquitin levels in an activity-dependent manner [66].

Another pivotal signaling pathway involved in neuronal survival and synaptic plasticity is the Brain-Derived Neurotrophic Factor/tropomyosin receptor kinase B (BDNF/TrkB) signaling [67]. Indeed, the Aβ-induced alteration of BDNF-mediated trafficking is mimicked by the use of a pharmacological inhibitor of UCH-L1 and, notably, the use of TAT-linked UCH-L1 restores BDNF/ TrkB cascade in neurons [42]. Moreover, Guo and colleagues further investigated the impact of UCH-L1 on this cascade, specifically on TrkB, evincing UCH-L1’s role in the ubiquitination of TrkB and UCH-L1’s involvement in the promotion of hippocampus-dependent memory. The blockage of the UCH-L1-regulated deubiquitination of TrkB evokes a consequent decrease in TrkB activation [68]. This is true not only in the CNS, but also peripherally: Chen and colleagues depicted a crucial effect of UCH-L1 in neuromuscular junctions (NMJs), with UCH-L1 knockout mice presenting the dysregulated function of NMJs. This leads to structural defects and the progressive degeneration of the motor nerves, suggesting that UCH-L1 is required for NMJ homeostasis in mice too [69].

An overview of the experimental approaches and major findings linking UCH-L1 to Alzheimer’s disease pathogenesis across different model systems is presented in Table 1. The schematic representation of the main pathways involved is illustrated in Figure 3.

## 3. UCH-L1 as a Target and Modulator of Neuroinflammatory Pathways

The term neuroinflammation refers to the activation of the CNS’s innate immune system in response to injury, infection, or the accumulation of abnormal proteins [70]. Neuroinflammatory responses are mediated by the production of pro-inflammatory cytokines, such as IL-1β, IL-6, and TNFα, chemokines, and reactive oxygen species (ROS) [71]. Most of these mediators are produced by the immune cells resident in the brain and spinal cord, namely astroglia and microglia [72]; nevertheless, it is now evident that the increased permeability of the blood–brain barrier (BBB) determines peripheral leukocyte infiltration, which contributes to chronic inflammation too [73]. Peripheral immune cells can be detected in the CNS in many pathological conditions, including AD, PD, amyotrophic lateral sclerosis (ALS), and Huntington’s disease [74,75]. All of these neurodegenerative disorders are characterized by the accumulation of aggregation-prone proteins, which determines the loss of proteostasis and increased neuronal death [76]. The ubiquitin–proteosome system (UPS) is one of the main systems which grants protein homeostasis in the brain [77]. Interestingly, the loss of proteostasis and the consequent increase in toxic aggregates is directly associated with increased levels of activation of inflammatory cells in the brain [78,79]. In turn, high levels of inflammation alter the efficiency of all of those internal systems through which toxic protein aggregation is prevented [80,81], thus highlighting the detrimental chronic vicious cycle in which the immune cells in the brain and the molecules involved in protein clearance, including UCH-L1, are crucial players.

In particular, UCH-L1 engages in a complex and multifaceted interplay with neuroinflammatory pathways, including oxidative stress and ROS production: indeed, recent studies position UCH-L1 as a key scavenger molecule for ROS in the CNS, and this is crucial for the redox balance in neurons, resulting in its abundance in this cell type [82]. For instance, in the presence of mild oxidative conditions, cysteine residue 152 (Cys-152) is quickly oxidized, thus protecting another cysteine residue (Cys-90), whose modification would result in irreversible structural abnormalities for UCH-L1 [83]. Moreover, the enzyme methionine sulfoxide reductase A (MsrA) seems to be crucial in binding ROS to UCH-L1, thus reducing the amount of free radicals and preventing neuronal damage. Interestingly, MsrA expression is known to be significantly downregulated in the context of neurodegenerative disorders, including AD [84]. Despite its putative antioxidant role, UCH-L1 remains one of the main targets of oxidative damage; in 2004, Choi and colleagues showed that the massive oxidation of five methionine residues (Met-1, Met-6, Met-12, Met-124, and Met-179) as well as at a single cysteine site (Cys-220) eventually results in the irreversible alteration of UCH-L1 physiological activities, and these events have been described both in AD and PD post-mortem brains [21]. The abnormal oxidation of the UCH-L1 component determines an impairment in the function of the whole proteasome system, eventually resulting in the accumulation of damaged proteins and toxic aggregate formation [85]. As expected, lower levels of UCH-L1 have been described in the cerebral cortex of post-mortem brains of AD patients and this downregulation seems to be mediated by an Aβ-42 dependent increase in oxidative stress levels [41]. Interestingly, oxidized UCH-L1 tends to aggregate as well, becoming part of the very same protein aggregates it should remove with its physiological activity, like Lewy bodies in PD [12].

The UCH-L1-related dysfunctions play a crucial role in the exacerbation of neuroinflammation, but neuroinflammatory-typical signaling pathways seem to also have a direct effect on UCH-L1. The different pathological pathways involving inflammatory processes and protein degradation’s dysfunction seem to converge on NF-κB, a well-known transcriptional factor, whose nuclear translocation determines the expression of genes involved in cell survival and growth, stress responses, as well as inflammatory processes [86]. Our research group found that NF-κB determines not only an increase in BACE1 expression, but also the downregulation of UCH-L1 transcription. As a consequence, on the one hand, there is an increase in amyloid burden, on the other, the total impossibility to remove misfolded proteins. By contrast, when NF-κB is inhibited, NF-κB subunit p65 cannot enter the nucleus, thus preventing the transcription of inflammation-related genes and totally rescuing the UCH-L1 physiological levels [41]. The strict interaction between UCH-L1 and NF-κB is further proved by the fact that the UCH-L1 gene promoter contains NF-κB binding sites: when the transcription factor is translocated in the nucleus, it downregulates UCH-L1 expression, thus confirming the negative effect that NF-κB in particular, and neuroinflammation more generally, have on UCH-L1 expression [50] (Figure 3). The same feed-forward loop has also been described in a mouse model of ischemic injury, thus highlighting the role of NF-κB as a promoter of neuroinflammation as well as UCH-L1 downregulation-dependent protein aggregation in response to stress [49]. However, new evidence seems to suggest the opposite relationship, with UCH-L1 inhibition which is able to prevent NF-κB translocation in the nucleus even in the presence of a pro-inflammatory stimulus [87]. In this case, it is not neuroinflammation and neuroinflammation-related pathways that regulate proteasome system activity, but it is the proteasome system itself, through its subunit UCH-L1, that plays a crucial role in cytokine production. This idea is supported by Liang et al., as they also proved that the inhibition of UCH-L1 ultimately results in the impairment of NLRP3 inflammasome assembly, determining in this way lower levels of IL1β production [88]. This is true especially in microglia cells, whose main functions are phagocytosis and pro-inflammatory cytokine production upon activation [89]. As for microglia cells, we proved that they are not insensitive to NF-κB activation, indeed, following hypoxia or Aβ42 injection, there is not only a decrease in UCH-L1 expression, as already proved, but also a decrease in triggering receptor expressed on myeloid cells 2 (TREM2) expression, which strictly correlates with increased levels of cytokine production. IL-6 and TNF-α levels as well as TREM2 expression are completely rescued when UCH-L1 is rescued too [90].

Notably, the so-called “neurodegenerative microglia” are characterized by the secretion of high levels of extracellular vesicles (EVs) carrying different types of cargos, including phosphorylated tau, lipids, and microRNAs [91,92]. EVs themselves suggest a different role of UCH-L1 in microglia, shifting from an active functional player to an alternative signature marker. Indeed, specific microglia-derived EVs have been shown to co-express CX3CR1, the Fractalkine receptor typically expressed on exosome membranes, and UCH-L1. Interestingly, increased levels of CX3CR1+/UCH-L1+ EVs have been detected in the plasma of patients across different cohorts affected by multiple sclerosis (MS) and AD [93]. These results offer a peripheral readout of microglia activation in the brain, with a pivotal centrality of UCH-L1. Nevertheless, the small abundance of UCH-L1 in microglia cells when compared to neurons makes the investigation of the effect of this molecule on microglia-dependent inflammatory processes difficult and with controversial results; nevertheless it is evident that a connection exists and deserves further investigations.

The effect of UCH-L1 is not limited to microglial cells, astrocytes also appear to be influenced by its modulation, although through partially distinct mechanisms. Astrocytic cells are traditionally considered supportive cells, nevertheless they are now recognized as active players in neuroinflammatory and neurodegenerative processes [94]. Indeed, in the case of spinal cord damage, the activation of reactive astrocytes suppresses neural stem cell (NSC) activation through UCH-L1 inhibition. As expected, the proteasome compromission also increases protein aggregates at the site of injury, thus contributing to further levels of inflammation [95]. In the very same study, they also proved the crucial role of UCH-L1 in supporting NSC endogenous proliferation, thus consolidating the relevance of this protein in supporting tissue regeneration and dampening inflammation levels. This leads to the awareness that alterations in UCH-L1 expression and function should be further investigated to better dissect the role of protein aggregation and inflammation in the context of CNS lesions and injuries as well as in neurodegenerative diseases.

As mentioned before, neuroinflammation is the result of microglia and astroglia cell activation, as well as peripheral leukocyte infiltration through disrupted BBB. Indeed, the increased permeabilization of the BBB is a crucial event in neurodegenerative disorders, as it further amplifies neuroinflammatory events [75]. Although a direct role for UCH-L1 in BBB integrity has not been ruled out yet, it is known that neuronal injury-induced BBB leakage leads to a massive increase in peripheral UCH-L1 levels [96]. This could justify why UCH-L1 is a well-established marker for mild traumatic brain injury (mTBI), where UCH-L1 levels are higher [97]. Moreover, UCH-L1 has been shown to have an important contribution to blood–spinal cord barrier repair after spinal cord injury (SCI). UCH-L1 stabilizes Sox17, an endothelial cell-specific transcription factor, which is critical for endothelial development [98]. In turn, Sox17 is a positive inducer of Wnt/β-catenin signaling, which participates in the determination and maintenance of BBB features [99]. As a consequence, the UCH-L1-driven tuning of Sox17 may represent a promising strategy through which the BBB permeability could be controlled under pathological conditions, thus contributing to the integrity and protection of the whole CNS.

Overall, these findings demonstrate that UCH-L1 is critically involved in multiple pathways underlying AD pathogenesis, including amyloid and tau metabolism, synaptic function, and neuroinflammatory responses. A concise overview of these mechanisms is provided in Table 2.

## 4. UCH-L1 as a Predictive Cerebrospinal Fluid (CSF) Biomarker for AD

Given the centrality of different hallmarks associated with AD and their pathogenic relevance in disease onset and progression, recent research has focused on translating these insights into measurable and predictive biomarkers for the diagnosis of AD and related dementia, i.e., p-tau181, p-tau217, and p-tau231 [100]. Given its pathological alterations in neurodegenerative conditions, UCH-L1 has gradually gained attention as a potential biomarker. For instance, several studies have investigated UCH-L1 levels both centrally, in particular in the cerebrospinal fluid (CSF), and systemically, in the plasma, considering AD cohorts with varying degrees of cognitive decline and comparing their findings with other clinically well-recognized biomarkers [101,102,103,104,105,106,107,108,109].

The first study was conducted by Öhrfelt and colleagues, who investigated UCH-L1 levels in the CSF. They reported a strong increase in the levels of UCH-L1 in AD patients from two different independent cohorts, with no differences between other dementia or sMCI (stable Mild Cognitive Impairment). This suggests that UCH-L1 dysfunction may occur already in the earlier preclinical phases of the disease. Intriguingly, UCH-L1 levels in CSF can discriminate AD patients from controls with high diagnostic accuracy, as suggested by an ROC (Receiver Operating Characteristic) curve analysis. The increase in UCH-L1 is paired with the parallel increase in p-tau 231, supporting its role as an additional predictive biomarker for AD [101]. Other data confirmed that UCH-L1 concentration increases in CSF and found a negative correlation with the MMSE (Mini Mental State Examination) score, a validated measure of cognitive function. This suggests that higher UCH-L1 levels are associated with a significant worsening of cognitive impairment [102]. Consistently with these observations, proteomic analyses carried out by several research groups identified an AD-driven increase in UCH-L1 in the CSF, with no statistically significant increase observed in other non-AD neurological diseases [103], or in a combined groups of participants with MCI or AD diagnoses [104]. Comparable evidence was obtained with SIMOA (Single Molecule Array), a routinely and well-recognized immunoassay technology based on the ultrasensitive detection of molecules in different biological samples [110]. Chatziefstathiou and colleagues evaluated several potential predictable biomarkers in AD and frontal temporal dementia (FTD). They found that UCH-L1 increases in an AD clinical context, alongside other biomarkers, such as Glial Fibrillary Acidic Protein (GFAP) and Tau and Neurofilament Light Chain (NfL), in line with previous reports. Moreover, they further performed a sample stratification using Sparse Partial Least Squares Discriminant Analysis (sPLS-DA) and they proved that UCH-L1 levels measured not only in the CSF, but also in the plasma, were valid biomarkers for distinguishing between AD patients and controls, pointing to the relevance of this enzyme in the context of neurodegenerative disorders [105]. When patients were analyzed across the AD continuum, from preclinical AD to MCI-AD and fully developed AD dementia, CSF UCH-L1 levels increased progressively when compared with MCI and subjective cognitive decline (SCD), in the presence of negative CSF AD profiles. This increase correlated with other established indicators, such as Aβ42/40 ratio, phosphorylated tau, total tau, and MMSE score [106]. Overall, UCH-L1 has emerged as a reliable early-stage biomarker in CSF. Specifically, its utility in distinguishing early-stage groups (pre-AD and MCI-AD) is evident from studies applying the A/T/N classification scheme (amyloid (A), tau (T), and neurodegeneration (N)) [111]. High CSF UCH-L1 levels were associated with a more rapid transition from T- to T+ status, indicating that high levels of UCH-L1 are predictive at least of tau pathology in AD prodromal phases. The fact that its levels change rapidly in the early stages of the disease strengthens the fact that UCH-L1 can be considered an early biomarker for AD diagnosis [107]. In particular, its detection results corroborate and are supplementary to other well-known CSF biomarkers, such as p-tau181, which correlates well with amyloid/tau pathology [112], and p-tau217, which demonstrated a superior performance compared with p-tau181 in terms of sensitivity and specificity for AD [113,114]. Despite the promising results, the collection of CSF from patients is an invasive and poorly feasible approach from the perspective of using UCH-L1 as a routine biomarker for dementia diagnosis. For this reason, measurements of this enzyme in plasma have been proposed as a simpler and cost-effective alternative to the more invasive approach of CSF collection. Unfortunately, conflicting results have emerged on the potential use of UCH-L1 as a plasmatic biomarker for AD. Although plasma UCH-L1 levels have been found to correlate with the cognitive decline measured by MMSE scores, no significant differences were observed between patients with MCI or AD and healthy controls from plasma measurements. By contrast, elevated plasmatic UCH-L1 levels have been reported in individuals suffering from PD and dementia with Lewy bodies (DLB) [115], revealing that plasmatic levels of UCH-L1 are inconclusive for AD [108], but they may be used for other neurological condition diagnosis, such as Parkinson’s disease, DLB, epilepsy, and amyotrophic lateral sclerosis (ALS) [116,117,118,119]. Although its role as a plasma biomarker in AD remains unconfirmed, UCH-L1 detection in blood has been recognized since 2018 by the Food Drug Administration (FDA) as a biomarker for mild traumatic brain injury (mTBI) [97]. Although UCH-L1 is currently not considered a predictable plasmatic biomarker for AD, conditions associated with dysregulated UCH-L1 levels could potentially predispose individuals to the onset of AD. Indeed, recent works from LoBue and colleagues evaluated how the impact of a history of TBI influences serum biomarkers in diverse cohorts with normal cognition, mild cognitive impairment, or dementia. Contrarily to other reports, their results highlighted an increase in UCH-L1 serum levels in parallel with lower MMSE scores, and significantly higher levels when MCI and dementia groups are compared with controls [109]. No significant differences in serum UCH-L1 levels were observed between TBI-positive and TBI-negative participants and no direct correlation with symptom duration was found in patients with a history of TBI showing a distinct temporal trajectory. In the TBI+ group, UCH-L1 levels initially increased shortly after symptom onset, then normalized, and were followed by a delayed and massive increase at later stages. The authors speculated that the early increase might reflect an initial activation of cell recycling processes involving UCH-L1, aimed at counteracting TBI-related damage. Over time, however, this protective response may diminish, leading to similar biomarker levels as those without a TBI history. Eventually, this process could become dysregulated, resulting in a pronounced late-phase elevation of UCH-L1, which may contribute to an increased vulnerability to AD.

UCH-L1 may represent a promising biomarker for AD, especially in CSF. However, regarding its potential use as a plasma biomarker, current data do not support its clinical utility. Indeed, even if the collection of CSF is certainly more invasive, UCH-L1-derived levels probably better mirror what happens in the CNS; on the contrary, UCH-L1 peripheral expression suffers from a lack of knowledge about the mechanistic events which are responsible for the flux in the bloodstream of a protein which is almost only neuronal. Nevertheless, longitudinal studies and novel advanced technologies could better elucidate the role of UCH-L1 levels in plasma and improve its reliability in AD blood-based diagnostics.

## 5. Conclusions

UCH-L1 is one of the most abundant proteins expressed in the brain and its high levels reflect its central role in maintaining cellular homeostasis. In particular, UCH-L1 regulates ubiquitin levels and orchestrates protein degradation. Additionally, it interacts with various intracellular modulators, especially in the context of AD, including APP, tau protein, NFTs, and neural circuitries. Impaired UCH-L1 functionality, whether in activity, stability, or expression may contribute to the pathogenesis of dementia. From a translational perspective, UCH-L1 has also emerged as a promising early-stage biomarker, particularly in the CSF, where its levels correlate with disease progression, MMSE scores, and tau pathology. In line with the A/T/N classification framework [111], elevated CSF UCH-L1 has been associated with a rapid transition from tau-negative to tau-positive states, reinforcing its potential utility in identifying prodromal AD stages. However, its evaluation in peripheral blood remains inconclusive, as plasmatic levels do not consistently distinguish AD from other neurodegenerative conditions. Nonetheless, its validated role as a blood biomarker in mTBI and its involvement in chronic post-TBI trajectories suggest that it may act as a susceptibility factor in individuals at a higher risk for AD. Further research is needed to clarify the pathogenic role of UCH-L1 in AD. In particular, novel therapeutic strategies aimed at counteracting AD-associated UCH-L1 depletion may offer valuable tools for halting AD progression, which remains an incurable neurodegenerative disorder. Different therapeutic approaches have been proposed to promote UCH-L1 stability and, consequently, enhance its activity. For instance, TAT-UCH-L1 recombinant proteins, adeno-associated virus (AAV), and lentiviral vectors have been successfully employed in preclinical models of Alzheimer’s disease (AD) [24,39,40,42]. Several drugs, such as LDN-57,444, MT16-001, and IMP-1710, have been developed for targeting UCH-L1’s detrimental upregulation in a cancer context, where they act as UCH-L1 inhibitors [10]. However, currently there is no UCH-L1 activator that can selectively enhance UCH-L1 activity. Future research should focus on the identification and in-depth characterization of molecules capable of safely increasing UCH-L1 activity in the central nervous system. Importantly, such interventions should avoid stimulating systemic UCH-L1 activity, as this could trigger aberrant overactivation and potentially lead to undesired side effects.

## Figures and Tables

**Figure 1 ijms-26-09012-f001:**
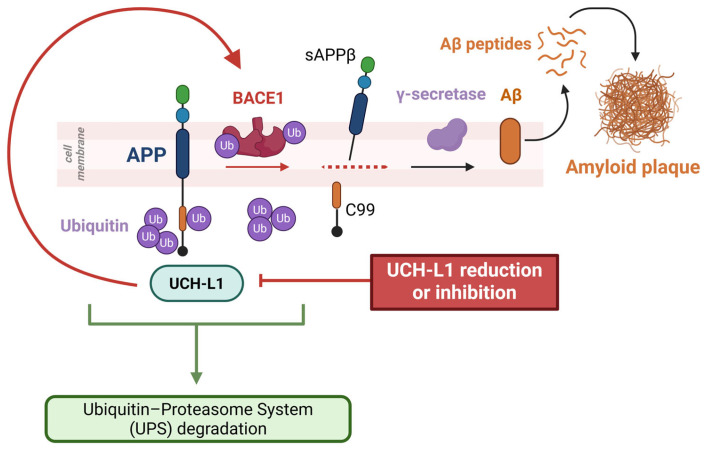
UCH-L1 plays a crucial role in the regulation of the pathogenic mechanisms that lead to Aβ plaque formation by a dual mechanism, with a direct and indirect effect on the amyloid precusor protein (APP) processing pathway. On one hand, it interacts with APP by promoting its ubiquitination, thus regulating APP turnover within neurons; on the other hand, it indirectly regulates APP processing by promoting BACE1 degradation. Reduced UCH-L1 expression results in higher levels of APP and BACE1 levels, thereby creating the ideal conditions for the generation of amyloid plaques (Created with BioRender.com, Created in BioRender. Guglielmotto, M. (2025) https://BioRender.com/tksi419 accessed on 20 July 2025).

**Figure 2 ijms-26-09012-f002:**
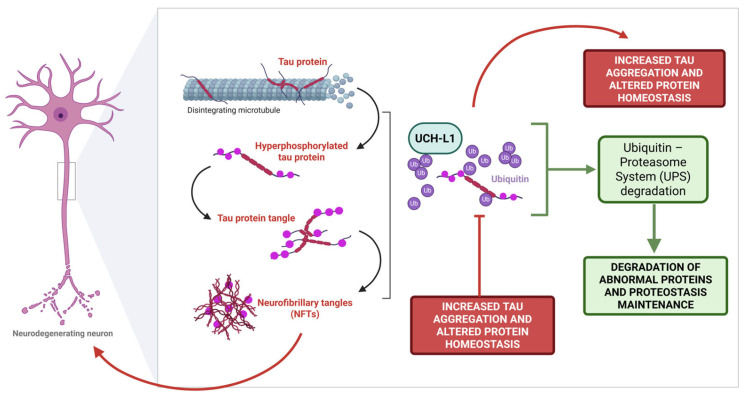
UCH-L1 strictly regulates protein homeostasis in the brain, including tau protein stability and phosphorylation levels. Under physiological conditions, UCH-L1 preserves tau solubility and prevents its pathological phosphorylation. In AD and other tauopathies, lower UCH-L1 levels contribute to protein misfolding, thus promoting the formation of NFTs, which ultimately lead to synaptic dysfunction and progressive neuronal loss (Created with BioRender.com, Guglielmotto, M. (2025) https://BioRender.com/egzowmu accessed on 20 July 2025).

**Figure 3 ijms-26-09012-f003:**
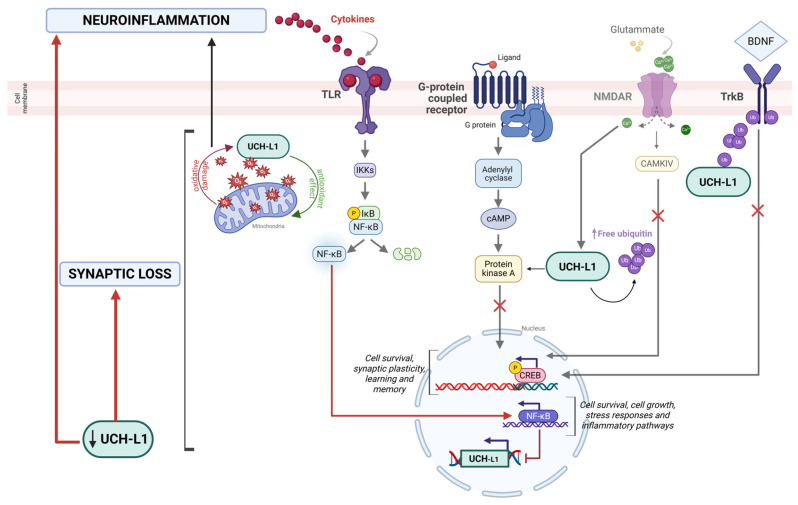
UCH-L1 is the most abundant protein in neuronal cells, hence the wide range of processes this protein is involved in, mainly affecting neuroinflammation and synaptic processes. Generally, NDMAR-driven stimulation of synaptic activity significantly upregulates UCH-L1 activity and promotes free monomeric ubiquitin, necessary for correct proteostasis. Indeed, UCH-L1 takes part in PKA-CREB signaling: rescuing UCH-L1 depletion repristinates PKA-regulatory subunit IIα, promoting CREB phosphorylation, and ultimately synaptic plasticity maintenance, learning, and memory. UCH-L1 also interacts with the BDNF/TrkB pathway. Indeed, UCH-L1 participates in the ubiquitination of TrkB and the rescue of UCH-L1 restores the BDNF/TrkB cascade, which allows the maintenance of neuronal functionality. In parallel, UCH-L1 is tightly linked with neuroinflammatory signaling. UCH-L1 interacts with the NF-κB pathway, supporting a feed-forward loop which exacerbates inflammatory processes. Oxidative stress, which closely relates to neuroinflammation too, is also a crucial event linked to UCH-L1: on one hand, this enzyme plays a key scavenger role, on the other hand, when reactive oxygen species (ROS) production increases, UCH-L1 activity is compromised by oxidative damage itself. (Created with BioRender.com, Guglielmotto, M. (2025) https://BioRender.com/klk3wji accessed on 20 July 2025).

**Table 1 ijms-26-09012-t001:** Experimental evidence of UCH-L1 involvement in Alzheimer’s disease pathogenesis. Summary of key findings from post-mortem studies, animal models, and cell-based systems supporting the role of UCH-L1 in amyloid-beta and tau metabolism and synaptic function.

Model/System	Experimental Approach	Findings
Human AD brain (post mortem)	Proteomic and ICC studies [9,18,21,41,42,43,58,61]	- ↓ soluble UCH-L1 in cortex/hippocampus - Redistribution in plaque-rich areas - Colocalization with NFTs
Knockout mice (gad)	UCH-L1-null mutation [44,65]	- ↑ Aβ accumulation, synaptic dysfunction, memory deficits
APP/PS1, APP23/PS45 mice	Overexpression or peptide restoration of UCH-L1 [39,40,49]	- ↓ plaque burden, rescued LTP and memory
Cell lines (HEK293, SH-SY5Y, N2a)	siRNA knockdown/overexpression [40,48,57,59,60]	- UCH-L1 loss → ↑ tau phosphorylation, aggregation - UCH-L1 gain → reduced Aβ and tau burden
Aplysia and hippocampal slices	UCH-L1 modulation [39,63,64]	- Regulates PKA–CREB, synaptic plasticity, and memory formation

The arrows provide a quick visual representation of the increase or decrease of the marker.

**Table 2 ijms-26-09012-t002:** Summary of the main roles of UCH-L1 in Alzheimer’s disease pathogenesis. The table highlights the contribution of UCH-L1 alterations to amyloid and tau pathology, synaptic dysfunction, and neuroinflammation/oxidative stress.

AD Hallmark/ Mechanism	Main Findings	Relevance to AD
Amyloid-β plaques	- UCH-L1 promotes APP degradation and regulates BACE1 stability [39,40,41,43,44,48,49] - ↓ UCH-L1 → ↑ Aβ accumulation; overexpression/TAT-UCH-L1 reduces plaque burden [39,40]	- Dysfunction of UCH-L1 accelerates amyloid pathology - Restoration counteracts Aβ load
Tau protein and NFTs	- Loss/inhibition of UCH-L1 → ↑ tau phosphorylation and aggregation [57,59,60] - UCH-L1 colocalizes with NFTs in AD brain [21,58,61] - Soluble UCH-L1 inversely correlates with NFTs [21]	- UCH-L1 maintains tau homeostasis - Dysfunction drives NFT accumulation
Synaptic dysfunction	- UCH-L1 regulates PKA–CREB [39,63,64,65] and BDNF/TrkB cascades [42,66,67,68] - Deficiency impairs LTP, memory, and plasticity; overexpression rescues synaptic function [39,40]	- Reduced UCH-L1 activity contributes to cognitive decline in AD
Neuroinflammation/ Oxidative stress	- UCH-L1 undergoes oxidation, nitrosylation, and carbonylation [20,21,22,23,24] - NF-κB suppresses UCH-L1 expression, fueling inflammation and amyloid accumulation [41,49,50,86] - UCH-L1 inhibition impairs NLRP3 inflammasome and cytokine production [87,88]	- Oxidative damage and inflammatory signaling impair UCH-L1 - Amplifying neurodegeneration

The arrows provide a quick visual representation of the increase or decrease of the marker.

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
