# Peer review of "UCH-L1 in Alzheimer’s Disease: A Crucial Player in Dementia-Associated Mechanisms"

_ijms, 2025, doi:10.3390/ijms26189012_

Round 1

Reviewer 1 Report

Comments and Suggestions for Authors

Dear Author,

The article is well-organized, with distinct sections and informative figures/tables, making it a valuable reference for understanding the diverse roles of UCH-L1 in Alzheimer’s disease

However, some areas could benefit from expanded discussion and clarification for broader impact.

Major Comments:

  • The section on neuroinflammation could be strengthened by discussing recent advances in UCH-L1's interaction with specific microglial phenotypes (e.g., disease-associated microglia) or its role in BBB integrity, as emerging studies suggest broader implications.
  • While the biomarker section is strong for CSF, expand on why plasma levels are inconsistent—perhaps include hypotheses on peripheral vs. central UCH-L1 dynamics or suggestions for future standardization (e.g., via SIMOA assays).
  • The conclusion mentions therapeutic potential but lacks specific examples (e.g., UCH-L1 restoratives like TAT-linked peptides). Suggest adding a brief outlook on drug development challenges.

Minor comments:

  • Line 95: Clarify the controversy around UCH-L1's ligase activity—cite additional references if needed to address ongoing debate.
  • Figures 1-3: Ensure consistent labeling (e.g., standardize arrow styles for inhibition/activation) for better visual clarity.

Author Response

REWIEVER 1

We appreciate the reviewer’s comment on the critical organisation of the manuscript and of the impact of our work in unravelling the role of UCH-L1 in in Alzheimer’s disease

Here we discuss the reviewer’s comments and suggestions point by point.

Major comments

  1. The section on neuroinflammation could be strengthened by discussing recent advances in UCH-L1's interaction with specific microglial phenotypes (e.g., disease-associated microglia) or its role in BBB integrity, as emerging studies suggest broader implications.

We thank the reviewer for the valuable suggestions. We conducted a thorough literature search on damaged associated microglia, and we found out interesting evidence about the upregulation of EVs production as a novel molecular signature for microglial pathological phenotype. In particular, microglia-derived EVs expressing UCH-L1 (CX3CR1+/UCH-L1+ EVs) are detected in different neurodegenerative disorders (multiple sclerosis and AD) [lines 391 – 400]. In parallel, we included a perspective on UCH-L1 indirect role in BBB integrity: UCH-L1 increase in the peripheric system has been widely documented after neuronal damage and we propose UCH-L1-dependent stabilisation of Sox17 factor in the orchestration of BBB architecture [lines 417 – 431].

  1. While the biomarker section is strong for CSF, expand on why plasma levels are inconsistent—perhaps include hypotheses on peripheral vs. central UCH-L1 dynamics or suggestions for future standardization (e.g., via SIMOA assays).

We sincerely thank the reviewer for the insightful remarks. We included, as suggested, at the end of the biomarker section our hypothesis about the possible reasons why the UCH-L1 data collected in CSF results more reliable and less controversial that the ones in plasma. In particular, we believe that one of the possible explanations could be found in the lack of mechanistic insights on UCH-L1 efflux from CNS to the bloodstream [lines 549 – 552].

  1. The conclusion mentions therapeutic potential but lacks specific examples (e.g., UCH-L1 restoratives like TAT-linked peptides). Suggest adding a brief outlook on drug development challenges.

We thank the reviewer for the valuable comment. We included in the conclusions [lines 575 – 585] a brief lookout on the available therapeutic strategies targeting UCH-L1, addressing some future perspective and challenges for the future development of UCH-L1 activator molecules, to modulate its activity at least in the CNS.

Minor comments:

Line 95: Clarify the controversy around UCH-L1's ligase activity—cite additional references if needed to address ongoing debate.

We thank the reviewer for the suggestion. We included in the text an additional reference on the still controversial dual role of UCH-L1 as a ligase, underlining the need of a better characterisation of this additional activity in vivo [lines 109 – 116].

Figures 1-3: Ensure consistent labeling (e.g., standardize arrow styles for inhibition/activation) for better visual clarity.

We thank the reviewer for the comment. We modified the labels of the Figures according to the request.

Reviewer 2 Report

Comments and Suggestions for Authors

Porchietto and colleagues conducted a thorough review on the connection between UCH-L1 and Alzheimer’s disease. This manuscript is timely and well-crafted, providing a detailed overview of UCH-L1 biology in AD. It skillfully combines mechanistic insights with its potential as a biomarker.

However, the manuscript would benefit from greater clarity, reduced redundancy, and stronger critical discussion of controversies and translational challenges. Below are detailed comments.

comments

  1. The introduction should better highlight how this review differs from previous UCH-L1/UPS reviews.
  2. The information on biomarkers is outdated. Provide current coverage of biomarker studies up to 2024.
  3. UCH-L1 can have both protective and harmful effects depending on modifications such as oxidation and nitrosylation. This dual nature acts as a “double-edged sword.”
  4. Determine whether UCH-L1 provides additional diagnostic information beyond pTau181/pTau217 or if it mainly functions as a supplementary marker.
  5. This review does not include a therapeutic discussion. Could targeting UCH-L1 activity for stability serve as a drug strategy, or could altering it lead to unforeseen effects due to its dual functions?
  6. Some aspects of the figures could be misleading or incomplete. Figure 1's legend is too brief, stating that UCH-L1 regulates APP and BACE1 turnover without explaining the dual mechanisms. In figure 2, the figure suggests that loss of UCH-L1 directly phosphorylates tau, but it actually does so by disrupting proteasomal clearance and aggresome function. In figure 3, the legend does not adequately reflect the complexity of the pathways depicted.

Author Response

We thank the reviewer’s comment on the overall structure of the manuscript, and the appreciation for the integrated aspects of both mechanistic insights of UCH-L1 in AD and its potential role as a biomarker. Here we discussed in detail its insightful comments and suggestions.

  1. The introduction should better highlight how this review differs from previous UCH-L1/UPS reviews.

We thank the reviewer for the suggestion. We have now included a brief comment on the novelty of our review, compared to the current state-of-the-art [lines 130 – 135].

  1. The information on biomarkers is outdated. Provide current coverage of biomarker studies up to 2024.

We thank the reviewer for the insightful comment. We acknowledge the importance of incorporating the most recent advancements in UCH-L1 as biomarker research in Alzheimer's disease. In our manuscript, we cited sources up to 2024, focusing specifically on studies related to Alzheimer's disease. The source that may be considered dated is from 2016 (https://doi.org/10.1159/000447239); however, we included this study because it was the first significant evidence in this area and laid the bases for subsequent research. We appreciate your feedback, and we are open to any further suggestion to enhance the relevance and accuracy of our manuscript.

  1. UCH-L1 can have both protective and harmful effects depending on modifications such as oxidation and nitrosylation. This dual nature acts as a “double-edged sword.”

We appreciate the reviewer’s remarkable contribution. We have now included an insight of the controversial role of these two UCH-L1 post-transcriptional modifications [lines 87 – 93]. We highlighted the idea that, it would lead to an impairment of UCH-L1 functionality, but the oxidated and nitrosylated isoforms could also represent two detectable variants to be targeted for dampening their detrimental effect in neurodegenerative disorders.  

  1. Determine whether UCH-L1 provides additional diagnostic information beyond pTau181/pTau217 or if it mainly functions as a supplementary marker.

We thank the reviewer for the suggestion. We have now better elucidated the supplementary role of UCH-L1 in CSF in addition to the detection of other well recognized CSF biomarker, such as p-tau217 and 218, as suggested by the reviewer [lines 510 – 514].

  1. This review does not include a therapeutic discussion. Could targeting UCH-L1 activity for stability serve as a drug strategy, or could altering it lead to unforeseen effects due to its dual functions?

We thank the reviewer for the valuable comments. We included the available therapeutic options targeting UCH-L1, addressing some questions for the future development of UCH-L1 activators in the CNS [lines 575 – 585].

  1. Some aspects of the figures could be misleading or incomplete. Figure 1's legend is too brief, stating that UCH-L1 regulates APP and BACE1 turnover without explaining the dual mechanisms. In figure 2, the figure suggests that loss of UCH-L1 directly phosphorylates tau, but it actually does so by disrupting proteasomal clearance and aggresome function. In figure 3, the legend does not adequately reflect the complexity of the pathways depicted.

We thank the reviewer for the comments. We have now modified the figures and the legends according to the suggestions.